# Immunogenic Properties and Antigenic Similarity of Virus-like Particles Derived from Human Polyomaviruses

**DOI:** 10.3390/ijms24054907

**Published:** 2023-03-03

**Authors:** Asta Lučiūnaitė, Indrė Dalgėdienė, Emilija Vasiliūnaitė, Milda Norkienė, Indrė Kučinskaitė-Kodzė, Aurelija Žvirblienė, Alma Gedvilaitė

**Affiliations:** Institute of Biotechnology, Life Sciences Center, Vilnius University, 7 Saulėtekio Ave, LT-10257 Vilnius, Lithuania

**Keywords:** polyomavirus, viral antigens, antibodies, immune response, macrophages

## Abstract

Polyomaviruses (PyVs) are highly prevalent in humans and animals. PyVs cause mild illness, however, they can also elicit severe diseases. Some PyVs are potentially zoonotic, such as simian virus 40 (SV40). However, data are still lacking about their biology, infectivity, and host interaction with different PyVs. We investigated the immunogenic properties of virus-like particles (VLPs) derived from viral protein 1 (VP1) of human PyVs. We immunised mice with recombinant HPyV VP1 VLPs mimicking the structure of viruses and compared their immunogenicity and cross-reactivity of antisera using a broad spectrum of VP1 VLPs derived from the PyVs of humans and animals. We demonstrated a strong immunogenicity of studied VLPs and a high degree of antigenic similarity between VP1 VLPs of different PyVs. PyV-specific monoclonal antibodies were generated and applied for investigation of VLPs phagocytosis. This study demonstrated that HPyV VLPs are highly immunogenic and interact with phagocytes. Data on the cross-reactivity of VP1 VLP-specific antisera revealed antigenic similarities among VP1 VLPs of particular human and animal PyVs and suggested possible cross-immunity. As the VP1 capsid protein is the major viral antigen involved in virus-host interaction, an approach based on the use of recombinant VLPs is relevant for studying PyV biology regarding PyV interaction with the host immune system.

## 1. Introduction

Polyomaviruses are non-enveloped dsDNA viruses of the *Polyomaviridae* family that infect birds, fish, and mammals, including humans [1,2]. The first representatives of this virus family in humans were John Cunningham (JC) and BK PyVs discovered in 1971 [3,4]. JCPyV can affect the brain, causing progressive multifocal leukoencephalopathy, while BKPyV can damage kidneys, induce nephropathy, and is one of the leading causes of kidney transplant failure. Both viruses enter the body via the respiratory system and can also cause mild respiratory infections [5]. Only later, starting in 2007, were the additional twelve human polyomaviruses (HPyVs) identified.

According to the World Health Organisation, 50–80% of the human population is infected by HPyVs during childhood [5]. In general, HPyVs cause asymptomatic or mild infection; thus, most individuals are capable of combating these viruses. However, occasionally PyVs remain dormant within the host after the infection by integrating viral genes into the human genome [6,7]. Immune system disorders can provoke virus transit from the latent to the acute phase. Therefore, HPyVs are particularly dangerous to immunocompromised patients [8,9]. In some cases, HPyVs induce changes in the infected cells of the host and might be tumorigenic [10]. PyV genomes were detected in diverse types of cancer, such as skin cancer [11], lung adenocarcinoma [12], osteosarcoma, and glioblastoma [13]. Next to cancer, HPyVs are implicated in other human diseases, such as trichodysplasia spinulosa, dyskeratotic dermatitis, and respiratory illnesses [14]. The latter evidence emphasises the importance of studying PyV biology and their interaction with the host immune system.

PyV genomes usually encode 5–9 proteins [15]. It is known that the viral products of viruses with a small genome are multifunctional. The transforming potential of PyVs is mostly related to the expression of the early non-structural proteins called tumor (T) antigens, which are involved in the regulation of viral transcription and genome replication but also have ability to inhibit tumour suppressing factors, such as the p53 and retinoblastoma protein, and by dysregulating cell signalling pathways can interfere with cell cycle regulation [10]. Nevertheless, other studies revealed an additional role of the major structural component of the viral capsid VP1 protein, showing that it also may participate in the regulation of cell cycle progression by interacting with microtubules and blocking the cell cycle at the G2/M phase [16].

An icosahedral capsid of PyVs is composed of 72 pentamers of major capsid protein VP1. Each pentamer is associated with the structural proteins VP2 and VP3. However, capsid-like structures can be assembled from VP1 alone and form VLPs [17,18]. VP1 VLPs mimic the actual size and shape of native viruses and may be useful in different studies of virus diagnostics, capsid structure, or virus entry pathways and cell tropism. It was demonstrated that recombinant HPyV-derived VP1 VLPs exhibited different hemagglutination activity (HA) that is related to virus-cell binding properties, which have an impact on viral spreading, cell tropism, and pathogenicity [18]. The capsid structure of PyVs is mostly determined by the amino acid sequence of VP1 protein which is responsible for binding to different sialic acid-containing receptors when entering the cell [19,20,21]. For instance, despite the similar structural properties of capsids, the tropism of JCPyV for glial cells is different from that of BKPyV, which points out that some structural differences and amino acid sequences of the VP1 protein might be related to completely different illnesses caused by JCPyV, BKPyV and other PyVs [22].

Previous studies suggest that PyVs are immunogenic [23]. Antibodies against the VP1 protein derived from different HPyVs were detected in human serum samples from various populations [24,25]. Our previous studies also demonstrated the immunogenic properties of recombinant JCPyV and BKPyV derived VP1 VLPs [17]. Furthermore, these investigations revealed some antigenic similarities between VP1 proteins derived from different PyVs. After immunisation with JCPyV VP1 VLPs, mouse and rabbit antiserum cross-reacted with BKPyV and SV40 VP1 VLPs, whereas mouse serum raised against BKPyV VP1 VLPs cross-reacted with JCPyV, SV40, budgerigar fledgling disease (BFPyV), hamster (HaPyV) and murine (MuPyV) PyV-derived VP1 VLPs [17].

Recombinant VLPs are already established in vaccinology and diagnostics as a safe alternative to intact viruses [26]. When used as adjuvants or antigen carriers in vaccines, VLPs can activate immune components. For example, it was reported that the ISCOMATRIX adjuvant induces inflammation [27]. Therefore, it is important to understand how different VLPs activate immune cells or whether their immunogenicity depends on their structural differences.

VLP structure is a crucial factor for inducing the immune response. Therefore, the immunogenicity of particular VLPs should be investigated in detail before their application as diagnostic tools or vaccine platforms. It was shown that the MuPyV VP1 capsid protein self-assembled to VLPs induced a 10–20-fold higher immune response compared to VP1 pentamers [28]. Another example, recombinant VLPs of JCPyV VP1 produced in the baculovirus expression system were morphologically similar to the JCPyV capsid, and showed high immunogenicity in rabbits when administered with an adjuvant. However, these VLPs without an adjuvant did not an induce immune response [29], in contrast to MuPyV [28].

Most of the studies on HPyVs biology usually involve investigations of one or a few representatives, but there are none that would exploit a wide range of them in one experimental design. In the present study, we explored the in house-developed collection of recombinant VP1 VLPs derived from different PyVs for the investigation of their immunogenic properties. The experimental data of antigenic similarities and differences among investigated PyVs derived VP1 VLPs was analysed, taking into account their phylogenetic relations based on the comparison of VP1 amino acid sequences. Furthermore, we have generated and characterised a panel of monoclonal antibodies against VP1 proteins derived from nine HPyVs and employed them for studying the uptake of corresponding VP1 VLPs by phagocytic cells.

## 2. Results

### 2.1. VP1-Derived VLPs of HPyVs Are Highly Immunogenic

We investigated the immunogenicity of recombinant yeast-produced VP1 VLPs derived from nine different HPyVs: JCPyV, KIPyV, WUPyV, MCPyV, HPyV6, HPyV7, TSPyV, HPyV9, MWPyV, and STLPyV (Table 1).

BALB/c mice were injected with the same doses (50 μg per mouse) of recombinant purified VLPs. The titres of antigen-specific IgG antibodies were measured in mouse antiserum samples by an indirect ELISA (Figure 1 and Appendix A).

After the first immunisation, the titres of antigen-specific IgG antibodies in four groups of mice immunised with VP1 VLPs of KIPyV, WUPyV, HPyV7, and TSPyV did not exceed 1:10,000. In mice immunised with MCPyV and HPyV9 VP1 VLPs, IgG titres exceeded 1:15,000, and in two groups of mice immunised with HPyV6 and STLPyV, VLPs were between those two titre values. After the second and third immunisations, the IgG titres strongly increased in all groups of mice except the groups immunised with HPyV7 and KIPyV VP1 VLPs. In addition, after the second or third immunisation, in seven groups out of nine, the IgG titres exceeded 1:15,000. Surprisingly, the group of mice immunised with HPyV7-derived VP1 VLPs did not elicit titres over the 1:10,000 value. The strongest antibody response was detected in mice immunised with HPyV9 VP1 VLPs: after the first immunisation, IgG titres reached 1:30,000, and after the second and third immunisations the IgG titres rose steadily up to 1:100,000.

We then proceeded with the immunogenicity analysis of HPyV-derived VP1 VLPs, assuming all immunisations as one unit. In this evaluation, we have compared the overall immune response to different VP1 VLPs and classified all VLPs into distinct groups from high to low immunogenicity (Figure 1 and Appendix A). HPyV9 VP1 VLPs induced the highest IgG response as compared to all other used VLPs. Another representative of highly immunogenic VLPs was MCPyV-derived VP1 VLPs. In contrast, VP1 VLPs of KIPyV and HPyV7 induced the lowest immune response. The VP1 VLPs of WUPyV, HPyV6 and TSPyV and STLPyV composed a group with a similar immune response pattern—lower titres after the first and second immunisation, but higher after the third immunisation. To summarize, the comparison of IgG titres showed that HPyV-derived VP1 VLPs are immunogenic but at distinct levels (Figure 1), and would fall in line from strongest to weakest as follows: HPyV9 > MCPyV > TSPyV, WUPyV, HPyV6, and STLPyV > JCPyV > KIPyV > HPyV7.

PyV-derived VP1 proteins used in our study self-assemble into VLPs of different sizes (Figure 2) [18]. Therefore, we analysed the immunogenicity data of VLPs in regard to their size. Only MCPyV and TSPyV VP1 proteins form large homogenous VLPs sized 40–50 nm in diameter. WUPyV, KIPyV, HPyV6 VP1 proteins form mostly smaller particles (25–35 nm), while HPyV9 and STLPyV VP1 proteins self-assemble into heterogeneous VLPs with a dominant size 40–50 nm in diameter and HPyV7 VP1 formed heterogeneous VLPs 30–40 nm in diameter. The smallest VLPs of WUPyV and HPyV6 induced a similar immune response with IgG titres ranging from 19,490 to 39,430 after the third immunisation, while KIPyV VP1 VLPs induced IgG titres barely exceeding 1:10,000, even after the third immunisation (Figure 1, Appendix A). The high levels of IgG response were induced by large heterogeneous and homogeneous VP1 VLPs of STLPyV, MCPyV, JCPyV and TSPyV. The antibody titre to heterogeneous HPyV7 VP1 VLPs sized 30–40 nm was lower than that induced by small KIPyV VP1 VLPs (25–35 nm in diameter). This data allowed us to assume that the smaller size of VP1 VLPs might have some negative impact on the humoral immune response, as overall regular VLPs 45–50 nm in diameter induced a better immune response than smaller VLPs of 25–30 nm in diameter.

After the immunogenicity study, the spleen cells of immunised mice were used for the generation of hybridomas producing monoclonal antibodies (MAbs) against recombinant VP1 VLPs of HPyVs from HPyV2 to HPyV9, and HPyV11. The list of MAbs is presented in Appendix A. The specificity of MAbs was investigated by ELISA and western blot. Some of the MAbs were used for immunocytochemistry analysis to investigate the uptake of VLPs by APCs (see below). These MAbs could be used as a highly specific tool for PyV detection and characterisation.

### 2.2. Antigenic Similarity of VP1-Derived VLPs of Different PyVs According to Serological Analysis

To evaluate the antigenic similarity of different human, mammalian and avian PyV VP1 VLPs, we analysed the cross-reactivity of polyclonal antibodies raised in mice against HPyV-derived VP1 VLPs. The phylogeny of the family *Polyomaviridae* is determined based on the amino acid sequences of the viral protein large tumour antigen (LTag). However, LTag-based phylogeny differs greatly from the phylogeny based on VP1 amino acid sequences [2]; thus, common PyV LTag-based classification into *Alpha*- through *Zeta*-polyomavirus is not suitable to compare VLPs of different PyVs, consisting of a single viral protein VP1. To compare the cross-reactivity data and the phylogenetic analysis of VP1, the phylogenetic analysis of VP1 amino acid sequences of VP1 VLPs used in this study was conducted (Figure 3A). According to this phylogenetic tree, all but the VP1 of SaraPyV (HPyV12 which was later recognised as *Sorex araneus polyomavirus 1*) representing a distinct single-species group, VP1 of PyVs, have cohered into five groups: *Wukipolyomavirus* (KIPyV, WUPyV, HPyV6, HPyV7, CVPyV), *Avipolyomavirus* (FPyV, BFDV), *Orthopolyomavirus II* (BKPyV, JCPyV, SV40) and *I* (MCPyV, TSPyV, HPyV9, NJPyV, LIPyV, HaPyV, MuPyV, YMPyV), and *Malawipolyomavirus* (MWPyV, STLPyV), corresponding to the data described previously [15,33]. The cross-reactivity data of polyclonal antibodies was analysed comparing the cross-reactivity among representatives of separate groups, as well as among VP1 VLPs of distinct PyVs. (Figure 3B).

The results of the cross-reactivity of antiserum, raised after immunisation with different VP1 VLPs, are in line with the phylogenetic analysis. After mouse immunisation with certain PyV VP1-derived VLPs, the cross-reactivity of antiserum was observed within the same VP1-derived group of *Wukipolyomavirus*, *Orthopolyomavirus I* and *II,* as well as *Malawipolyomavirus* of the *Polyomaviridae* family (Figure 3B). For example, VP1 VLPs of human *Wukipolyomaviruses* strongly reacted with mouse antisera raised within the group, except for non-human polyomavirus CVPyV. Furthermore, for all but the *Avipolyomavirus* group, non-human PyV VP1 VLPs showed lower cross-reactivity to HPyV VP1 VLPs derived antisera both within and outside their groups when compared to related HPyV VP1 VLPs, except the high cross-reactivity of VLPs derived from animal PyVs with MCPyV-derived VP1 VLP-specific antiserum (Figure 3B). On the other hand, avian PyV-derived VP1 VLPs showed strong reactivity with the antisera of mice immunized with HPyV-derived VP1 VLPs, with the exception of TSPyV-derived VP1 VLPs. Mouse antiserum raised against VP1 VLPs of JCPyV from the group *Orthopolyomavirus-II* showed strong cross-reactivity with VP1 VLPs of the same group, as well as moderate to strong cross-reactivity with VP1 VLPs of other groups. At large, antisera cross-reactivity among different PyV-derived VP1 VLP groups was widely observed. Antisera raised against MCPyV- and HPyV6-derived VP1 VLPs demonstrated the highest, and antisera raised against TSPyV- and HPyV9-derived VP1 VLPs displayed the lowest cross-reactivity with VP1 VLPs of other polyomaviruses. Mouse antiserum raised against STLPyV-derived VP1 VLPs also showed modest cross-reactivity with VP1 VLPs from the *Orthopolyomavirus I* and *Orthopolyomavirus II* groups; however, it strongly cross-reacted with the representatives of the *Malawipolyomavirus* group. Overall, the cross-reactivity data of VP1 VLP-specific antisera support the VP1 based PyV phylogeny of five distinct groups; however, high intergroup cross-reactivity was also observed.

### 2.3. The Uptake of Polyomavirus-Derived VP1 VLPs by Macrophages

To evaluate the ability of VP1-derived VLPs of different HPyVs to induce cellular immune responses including APC activation, we investigated whether these VLPs are taken up by professional APCs—macrophages. We treated primary cultures of murine macrophages with VP1 VLPs of nine HPyVs for 24 h and then observed their uptake by fluorescent microscopy using previously generated VP1 VLP-specific MAbs. To detect phagocytosis, the cells were simultaneously stained for lysosomal marker CD68. The observed co-localisation of VLPs with CD68 confirmed that macrophages interact with VLPs and engulf them (Figure 4). We showed that VP1 VLPs of all nine HPyVs were taken up by macrophages. Some differences in the intensity of the fluorescence signal for different VP1 VLPs were observed. However, they are not necessarily indicative for the differences in uptake efficiency of the used VP1 VLPs. The intensity of the fluorescence signal might be dependent on the affinity of the MAbs used for VP1 VLP labelling.

## 3. Discussion

Previous studies have suggested that VLPs can act as strong immunogens and can even can be exploited as carriers for target epitopes [34]. VLPs mimic the actual size and shape of native viruses and can mediate both humoral and cell-mediated immunity [35]. Because of their similarity to native viruses, VLPs are efficiently recognised by professional antigen presenting cells (APCs). The exposure of target antigens on the VLP surface allows antigen delivery and presentation to T and B cells, since the VLPs act as a delivery platform and an adjuvant at the same time. After the uptake of VLPs by APCs, they present the target antigens to naive T cells and initiate target-specific immune responses. However, it is important to select the appropriate conditions for a regular VLP assembly in order to get native-like particles, as their structural characteristics may impact their antigenic properties [36].

We demonstrated the high immunogenicity of different recombinant VLPs derived from the major capsid protein VP1 of nine HPyVs. Our data links with previous immunogenicity studies in humans or animals either naturally infected with PyVs or immunised with VLPs. The seroprevalence study of 14 HPyVs showed that antibodies against these viruses are detected in the human population, demonstrating that all these viruses are recognised by the human immune system [37]. Other studies demonstrated that BKPyV is able to induce IgG responses in infected individuals [38], and some other HPyVs are recognised by the immune system and induce humoral and cellular immune responses in humans [39]. On the other hand, some PyVs, such as SV40, can evade the immune system of the host [40]. The study on the immunogenicity of MuPyV VP1 VLPs in mouse models also demonstrated that these VLPs induced strong humoral and cellular immune responses [41]. In addition, our results show the impact of VLP size on humoral immune response, as larger VLPs induced a higher response. The differences in the humoral immune response could also be related to different amino acid sequences on the surface of VLPs that determine the exposure of different VLP epitopes to B lymphocytes.

Until now, only one or a few VP1 VLPs derived from different HPyVs were investigated within the same experimental setting. The uptake of PyV VP1-derived VLPs by the cells was also shown [28]. In contrast to previous studies, we demonstrated the uptake of VP1 VLPs derived from nine HPyVs by primary mouse macrophages. Previous studies suggest that PyVs enter the cells via endocytosis after VP1 protein binding to sialic acid-containing cellular receptors [42]. It was also demonstrated that PyV-derived recombinant VLPs are taken up by the cells in a comparable way as native viruses. For instance, VP1 VLPs of HaPyV were taken up by the dendritic cells and induced their activation [43]. Another example shows that VP1 VLPs of BK and JC PyVs were taken up by non-professional phagocytes—liver sinusoidal endothelial cells [44]. The uptake of VLPs by the cells is a fast process. For example, VP1 VLPs of the murine pneumotropic virus were significantly internalised within 10 min, and in 40 min a 50% uptake of these VLPs were detected in a murine endothelial cell line [45]. After endocytosis, VLPs were transferred to the cell nucleus by their nuclear localisation signal allocated in the N-terminal region of VP1 protein, as was demonstrated with VLPs of JCPyV using HeLa and SVG human cell lines [46]. We showed that HPyVs VLPs are recognised by both humoral and innate immune systems, suggesting that HPyVs itself might be immunogenic.

Certain PyVs are declared to be zoonotic, thus, we made a cross-reactivity analysis of different human, mammalian and avian PyV VP1 VLPs to identify possible antigenic differences between investigated PyVs. First, we made the phylogenetic tree of VP1 amino acid sequences of all VP1 VLPs used in this study. We set five distinct groups combining human, mammalian and avian PyVs VP1. Our cross-reactivity analysis of VP1 VLP-specific antisera was in agreement with the VP1 based PyV phylogenetic tree; however, a high intergroup cross-reactivity was also observed. It was previously shown that mouse polyclonal antibodies raised using HPyV VP1 VLPs are able to recognize other HPyVs VP1 proteins in western blot analysis [18], but it was a bit different from our data. It should be noted that, in this study, the antibodies were raised against the surface-exposed parts of VLPs, while the phylogenetic analysis is based on the comparison of the primary structure of VP1 proteins, which might cause some of the differences between phylogenetic and experimental data of antiserum cross-reactivity. We also found differences between other studies investigating VP1 amino acid sequences. For example, SaraPyV was shown to be similar to MCPyV, TSPyV, and HPyV9 [47], while our cross-reactivity data evince SaraPyV’s similarity to MCPyV, HPyV6, and STLPyV. These discrepancies could be due to different analysis methods, as we assess epitopes in the spatial structure of VP1, not just the amino acid sequence. It is likely that the structural antigenic comparison represents biological evidence of PyVs properties, especially when VLPs structures seem to be comparable to native viruses.

Recombinant VP1 VLPs of PyVs are extensively exploited as a promising tool for various biomedical applications. They are used for the investigation of PyV seroepidemiology [23], as they are excellent antigens in ELISA platforms to determine PyV-specific antibodies. Due to their repetitive structure and large size, VP1 VLPs of PyVs are highly immunogenic and can be used as carriers or as a delivery platform for the presentation of heterologous antigens in vaccine development [48]. These VLPs might be used to develop vaccines without additional adjuvants because PyV-derived VLPs are able to activate pattern-recognition receptors, thereby enhancing the immune response [23]. Technologies for constructing chimeric and pseudotype PyV-derived VLPs allow for the insertion of antigens of different sizes and origins into VLPs, which is a valuable approach for studies of non-immunogenic or difficult-to-express proteins [28]. On the other hand, PyV-derived chimeric VLPs harbouring other viral antigens as targets can be used to investigate the interaction at the cellular level between the target antigens and the host, for example, their binding with cell receptors and the mechanisms of virus uptake by the cells. Our study provides additional information about the immonogenicity of HPyVs VP1-derived VLPs, demonstrating their ability to induce immune response depending on their size, amino acid sequences, and other structural features.

Despite the many studies on PyV epidemiology and data about their presence in patients with certain diseases, there are insufficient data on the biology of these viruses and how the immune system combats them. Some HPyVs are dangerous; their activity might be disastrous to immunocompromised patients [5]. For example, JCPyV and BKPyV cause renal allograft failure after renal transplantation, and there is no effective treatment for this life-threatening condition [49]. Another example is skin diseases induced by HPyVs—MCPyV, TSPyV, HPyV7 and HPyV6 [50]. MCPyV stands out as an aggressive cancer-causing virus [51]. HPyV6 and HPyV7 are primarily associated with inflammatory skin diseases such as psoriasis [52]. Though some previous studies suggest that PyVs induce strong immune responses both at humoral and cellular levels, more investigations are needed on the interaction of PyVs with the host immune system and on PyV-driven molecular mechanisms to evade the immune responses, especially due to the fact that most individuals are infected with PyVs in their childhood, as highlighted by the recent study in Greece [53]. Understanding the risk factors for developing a severe PyV-caused disease is of high importance, as there is no specific treatment for PyV infection. The current treatments based on using neutralising antibodies or immunotherapy are not efficient sufficient and have side effects [54,55], highlighting the need for more studies in this field.

In this study we also generated a collection of MAbs of various specificities to HPyVs. This tool can be used for PyVs detection and characterisation. Generated VP1 specific MAbs could be applied for assays to detect the native viruses in cell cultures or tissues. MAbs’ specificity to a certain epitope may also be used to investigate the structure of viral particles and interactions between those particles and host cell receptors.

In sum, in this study we compared a number of VLPs composed of recombinant VP1 proteins of different human, mammalian and avian PyVs in regard to their immunogenicity, antigenic similarity, and their uptake by professional APCs. We demonstrated that despite the different size and some distinctions of immunogenicity levels, all nine studied HPyV derived VP1 VLPs were immunogenic in a mouse model and induced IgG formation. They were also taken up by mouse macrophages, suggesting the subsequent presentation of VLP-derived antigens for inducing a cell-mediated response. Antibodies raised against certain VP1 VLPs of HPyVs cross-reacted with VLPs of other human and animal PyVs. The pattern of cross-reactivity indicates the high antigenic similarity of PyV VP1 proteins, and is in agreement with their phylogenetic analysis based on the sequence of VP1 proteins. Furthermore, as the humoral immune response to PyVs is mainly related to VP1, the cross-reactivity levels between VP1 proteins may suggest possible cross-immunity to several PyVs after encountering one of them. More investigations are needed on the role and mechanisms of immune protection against severe diseases caused by different PyVs. In conclusion, our study and the developed molecular tools reveal the antigenic features of PyVs and provide the means for further studies of PyV biology.

## 4. Materials and Methods

### 4.1. Materials

Dulbecco’s Modified Eagle’s Medium (DMEM; cat#31966047), fetal bovine serum (FBS; cat#A3840402), penicillin/streptomycin (P/S; cat#15140122), Dulbecco’s Phosphate Buffered Saline (PBS; cat#14190250), Hanks’ Balanced Salt Solution (HBSS, cat#14170138), and 0.25% trypsin solution (cat#25050014) were obtained from Gibco, Thermo Fischer Scientific, Waltham, MA, USA. Cell culture plates: T75 culture flasks (cat#658170) were sourced from Greiner Bio-One, Kremsmünster, Austria; TPP multi-well tissue culture plates (cat#92012, cat#92024, cat#92048) came from TPP Techno Plastic Products AG, Trasadingen, Switzerland. ELISA micro test 96 well plates (cat#10-121-5100) came from Nerbe plus GmbH & Co. Poly-L-lysine (PLL; cat#1524), was obtained from Sigma-Aldrich by Merck, Darmstadt, Germany. Dimethylsulfoxide (DMSO; cat#A3672) was obtained from PanReac AppliChem and the ITW Reagents, Barcelona, Spain. Tween-20 (cat#9127.1) and sulphuric acid (H_2_SO_4_, cat#X873.1) came from Carl Roth, Karlsruhe, Germany. TMB Substrate ‘NeA-Blue’ (#01016-1) was sourced from Clinical Science Products, Inc., Mansfield, TX, USA. Secondary antibodies for ELISA and western blot Goat Anti-Mouse IgG (H + L)-HRP Conjugate (cat#1706516) came from Bio-Rad, Hercules, United States. Polyethylene glycol 1500 (PEG) was obtained from HybriMax, Sigma-Aldrich by Merck, Darmstadt, Germany. Hypoxantine, aminopterin and the thymidine (HAT) media supplement was from Sigma-Aldrich by Merck, Darmstadt, Germany. FBS for hybridoma came from Biochrom, Cambridge, United Kingdom. The monoclonal Antibody Isotyping Kit I (HRP/ABTS) was obtained from Pierce Biotechnology, Waltham, MA, USA. Paraformaldehyde (PFA; cat#158127) and Triton X-100 solution (cat#X100) were from Sigma-Aldrich by Merck, Darmstadt, Germany. Secondary antibodies: Donkey anti-Rat IgG (H + L) Alexa Fluor 488 (cat#A21208), Donkey anti-Mouse IgG (H + L) Alexa Fluor 594 (cat#A21203), nuclear stain Hoechst33342 (cat#H3570) were obtained from Invitrogen, Thermo Fisher Scientific, Waltham, MA, USA. PageBlue Protein Staining Solution (cat# 24620) was from Thermo Fisher Scientific, Waltham, MA, USA. Roti^®^-PVDF membrane 0.45 µm pore sizes (cat#T830.1) and powdered milk (cat#T145.1) were sourced from Carl Roth, Karlsruhe, Germany. The 4-chloro-1-naphtol and H_2_O_2_ (cat#C6788) came from Fluka, Sigma-Aldrich by Merck, Darmstadt, Germany.

### 4.2. Purification of Recombinant PyV-Derived VP1 VLPs from Yeast Cells and Their Analysis by Electron Microscopy

The recombinant PyV-derived VP1 VLPs used in this study (Table 1) were generated in yeast, purified, and analysed by electron microscopy as described previously [18]. Briefly, *Saccharomyces cerevisiae* yeast biomass harbouring recombinant proteins was mechanically disrupted in a DB450 buffer (450 mM NaCl, 1 mM CaCl_2_, 0.25 M L-Arginine and 0.001% Trition X-100 in 10 mM Tris/HCl-buffer, pH 7.2) with 2 mM PMSF and EDTA-free Complete Protease Inhibitors Cocktail tablets (Roche Diagnostics, Mannheim, Germany), and its clarified supernatant was transferred onto 30–60% sucrose gradient. After overnight centrifugation (at 4 °C) at 100,000× *g* (Beckman Coulter Optima L-90 ultracentrifuge), the collected fractions were analysed by SDS-PAGE. The mixture of fractions containing PyV VP1 proteins diluted in DB150 buffer (150 mM NaCl, 1 mM CaCl_2_, 0.25 M L-Arginine and 0.001% Trition X-100 in 10 mM Tris/HCl-buffer, pH 7.2) was subjected to ultracentrifugation at 100,000× *g* for 4 h at 4 °C. Afterwards, pellets containing VP1 were dissolved in a DB150 buffer and subjected to ultracentrifugation overnight on a CsCl gradient (1.23–1.46 g/mL density) at 4 °C. One millilitre fractions of the formed gradient were collected and analysed by SDS-PAGE. Positive fractions were pooled and additionally ultracentrifugated on a CsCl gradient overnight, as described above. VP1-derived VLPs were then precipitated by ultracentrifugation for 4 h at 100,000× *g* after pooling and diluting purified VP1-containing fractions in DB150 buffer. The precipitated VP1-derived VLPs were dissolved in PBS, dialysed against PBS, aliquoted and lyophilised or mixed with 50% glycerol for storage at −20 °C. The VLP formation was verified by the examination of the purified proteins by a Morgagni-268 electron microscope (FEI, Inc., Hillsboro, OR, USA), and the VLP stability was evaluated by a nanoparticle tracking analysis performed with a NanoSight LM10-HS (NanoSight, Amesbury, UK). The protein samples for electron microscopy were placed on 400-mesh carbon-coated palladium grids (Agar Scientific, Stansded, UK) and stained with 2% aqueous uranyl acetate.

### 4.3. Phylogenetic Analysis of PyVs Based on VP1 Protein Sequence

The unrooted phylogenetic tree was created based on PyV major capsid protein VP1 amino acid sequence analysis. The VP1 amino acid sequences of 21 PyVs were taken from the GenBank database (the names of virus species and GenBank accession numbers are presented in Table 1). VP1 amino acid sequences of those PyVs were aligned using MAFFT v7.475 L-INS-i algorithm [56]. The phylogenetic tree was generated using the raxmlGUI 2.0 program (version 2.0.5) [57]. Optimal model parameters (LG + F + I + G) were selected with in-built ModelTest-NG (version 1.0.1) [58]. A RaxML-NG [59,60] maximum likelihood analysis was run with 20 random starting trees and the autoMRE boot stopping option (cut-off 0.03). The resulting trees were observed and edited using iTOL [61].

### 4.4. Immunisation of Mice and Generation of Monoclonal Antibodies

BALB/c mice were bred and maintained in an animal facility at the Department of Immunology of the Centre for Innovative Medicine (Vilnius, Lithuania). The groups of three female mice aged 6–8 weeks per each antigen were immunised with PyV-derived recombinant VLPs. All injections were subcutaneous. The dose was 50 μg of protein per mouse. For the primary immunisations, the antigens were emulsified in complete Freund’s adjuvant (Sigma-Aldrich, St. Louis, MI, USA). The second immunisation followed on day 28, with antigens emulsified in incomplete Freund’s adjuvant. The third immunisation followed on day 56, with the antigens dissolved in PBS. Antiserum samples were collected on day 14 after the first, second, and third immunisations, and tested by an indirect enzyme-linked immunosorbent assay (ELISA) for the presence of IgG antibodies specific to PyVs. The spleen cells of the mouse with the highest antibody titre were used for the generation of hybridomas, as previously described [62]. Briefly, three days after the boost immunisation, the spleen cells of the mouse were fused with Sp2/0-Ag14 mouse myeloma cells using polyethylene glycol 1500 (PEG/DMSO solution, HybriMax, Sigma-Aldrich, Darmstadt, Germany). Hybrid cells were selected in growth medium supplemented with hypoxantine, aminopterin and thymidine (50× HAT media supplement, Sigma-Aldrich, Darmstadt, Germany). Samples of the supernatant from wells with viable clones were screened by an indirect ELISA. Hybridomas secreting polyomavirus specific antibodies were subcloned by a limiting dilution method. Hybridoma cells were maintained in complete Dulbecco’s Modified Eagle’s Medium (DMEM, Biochrom, Berlin, Germany) containing 15% fetal bovine serum (Biochrom) and 300 μg/mL gentamicin. Antibodies were isotyped using a Monoclonal Antibody Isotyping Kit I (HRP/ABTS) (Pierce Biotechnology, Rockford, IL, USA) in accordance with the manufacturer’s protocol. All procedures involving experimental mice were performed under controlled laboratory conditions in strict accordance with Lithuanian and European legislation. The immunisation was performed at The Center for Innovative Medicine (Vilnius, Lithuania), which has State Food and Veterinary Service permissions for breeding experimental mice (license No. LT 59-902, permission No. 184), and their use for the generation of polyclonal and monoclonal antibodies (permission No. 209).

### 4.5. Preparation of Mouse Antiserum

Antiserum was collected from the chest cavity of immunised mice after their sacrifice. The collected blood was mixed with PBS in a ratio of 1:1 and centrifuged at 300× *g* for 20 min. The supernatant was transferred to a new tube, and a saturated ammonium sulphate ((NH_4_)_2_SO_4_) was added in a ratio of 1:1 to precipitate proteins of higher molecular weight including immunoglobulins. After overnight incubation at 4 °C, the prepared samples of polyclonal antibodies were centrifuged at 12,000× *g* for 10 min. The precipitation was resuspended in PBS and precipitated again using ammonium sulphate in a ratio of 1:1. The prepared samples of polyclonal antibodies were stored at 4 °C.

### 4.6. Indirect Enzyme-Linked Immunosorbent Assay (ELISA)

Microtiter plates (Nerbe Plus GmbH, Winsen/Luhe, Germany) were coated with 100 μL/well of PyV VLPs diluted in the coating buffer (0.05 M sodium carbonate, pH 9.5) at a concentration of 5 μg/mL. The plates were incubated overnight at 4 °C. The coated plates were blocked with 250 μL/well of PBS containing 2% BSA for 1 h at room temperature (RT). The plates were then rinsed twice with PBST (PBS with 0.1 % Tween-20). Antiserum samples, hybridoma growth medium, or PAb were diluted in PBST, added to the wells (100 μL/well), and incubated for 1 h at RT. The plates were then incubated for 1 h with Goat Anti-Mouse IgG (H + L)-HRP Conjugate (Bio-Rad, Hercules, CA, USA) diluted 1:5000 in PBST. The enzymatic reaction was visualised by the addition of 100 μL of “NeA-Blue” TMB solution (Clinical Science Products, Mansfield, MA, USA) to each well. The reaction was stopped by adding 50 μL/well of 1 M sulphuric acid solution. The optical density (OD) was measured at 450 nm (reference filter 620 nm) with a microplate reader (Sunrise Tecan, Männedorf, Switzerland).

### 4.7. The Study of Immunogenicity and Antigenic Properties of VP1 VLPs

Antiserum samples collected from immunised mice were used for the immunogenicity study. Equal dilution factors of polyclonal antibodies were used to generate titration curves. Titres were determined with GraphPad Prism 9.2.0 (GraphPad Software, Inc., La Jolla, CA, USA) by applying the Sigmoidal, 4PL, X is log(concentration) model and calculating EC50 (half maximal effective concentration). For each investigated antigen, three biological experiments were used (polyclonal antibodies from three mice). For each antigen, the first, second, and third immunisations were grouped as one data set. The antibody titres data sets were compared with each other using a two-way ANOVA. The antigenicity refers to the antibody specificity to antigen. It was investigated by analysing the antibodies’ cross reactivity with antigens of different PyV VP1. Equal concentrations (µg/mL) of polyclonal antibodies were used to generate titration curves. Antiserum concentration was equated to the total protein concentration. Titres were determined with GraphPad Prism 9.2.0 (GraphPad Software, Inc., La Jolla, CA, USA) by applying the Sigmoidal, 4PL, X is log(concentration) model and calculating EC50. For each investigated antibody, two technical replicates were used. The calculated titres were laid out in the heatmap for comparison of each polyclonal antibody reactivity with different VP1 VLPs of human and animal PyVs.

### 4.8. Preparation and Treatment of a Primary Mouse Macrophage Culture

The primary cell cultures of mouse brain macrophages (microglia), were prepared as described previously [63]. Procedures with C57BL/6 mice were performed at the Vilnius University Life Science Center animal facility (Vilnius, Lithuania) in accordance with EU legislation (State Food and Veterinary Service permission No. LT 61-13-004). The pups were sacrificed by decapitation. The brains were separated from the cranium of neonatal 0- to 3-day old mice pups. Briefly, brains were stripped of the meninges and dissociated using mechanical shearing and trypsin. The cells of two brains were plated on one PLL-coated T75 culture flask (Greiner Bio-One, Kremsmünster, Austria) and cultivated in DMEM supplemented with 10% heat-inactivated FBS and 1% P/S. On the next day, cells were washed three times with PBS to remove cellular debris and cultured with DMEM supplemented with 10% FBS, 100 U/mL P/S, and 10% L929 conditioned medium as a source of growth factors. After approximately 8 days, loosely attached mature microglia were shaken off the astrocytic monolayer with a repetition of the harvesting procedure at every 2–3 days as many as three times. Cells were seeded at a density of 1 × 10^6^/cm^2^ in 1/2 old medium (condition medium from microglia shake) and 1/2 fresh DMEM supplemented with 10% FBS and 1% P/S, and allowed to adhere overnight. On the next day, microglia were washed with serum-free DMEM and treated with viral proteins (20 µg/mL) for 24 h in a serum-free DMEM containing 1% P/S. After 24 h of incubation, the cells were assessed by immunocytochemistry.

### 4.9. SDS-PAGE and Western Blot Analysis

To evaluate the interaction of the MAbs with denaturated VP1 antigens, a Western blot analysis (WB) was performed. The samples of recombinant polyomavirus proteins were boiled in a reducing sample buffer and separated in 4–12% polyacrylamide gel electrophoresis (PAGE) in an SDS-Tris-glycine buffer. Proteins were visualised by staining with PageBlue Protein Staining Solution (Sigma-Aldrich, Darmstadt, Germany). The proteins from the unstained SDS-PAGE gel were blotted onto Roti^®^-PVDF membrane (Carl Roth, Karlsruhe, Germany) by semidry electro-transfer. The membrane was blocked with 5% milk in PBS for 2 h at RT and rinsed with PBST. The membrane was then incubated with primary antibodies in PBST with 2% milk for 1 h at RT. Thereafter, the membrane was incubated with secondary antibodies Goat Anti-Mouse IgG (H + L)-HRP Conjugate (Bio-Rad, Hercules, CA, United States) diluted 1:4000 in PBST with 2% milk powder for 1 h at RT. The enzymatic reaction was developed using 4-chloro-1-naphthol and H_2_O_2_ (Fluka, Milwaukee, WI, USA) solution. For the analysis of monoclonal antibodies, undiluted hybridoma supernatants were used.

### 4.10. Immunocytochemistry for Studying the Uptake of VLPs by Mouse Macrophages

Cells were stained in TPP 24 well plates. After the treatment, the cells were washed with PBS and fixed in 4% PFA dissolved in PBS for 15 min and permeabilised with 0.1% Triton X-100 prepared in PBS for 10 min. A blocking solution consisting of PBS containing 2% BSA was applied for 30 min, followed by two washing steps. The primary antibodies rat anti-CD68 (1:1000; Clone FA-11) and mouse anti-PyV VLPs (MAbs of hybridoma supernatant at dilution 1:2) were added to the blocking solution and incubated overnight. Goat anti-rat (1:1000) and Goat anti-mouse (1:1000) secondary antibodies were used, respectively. The secondary antibodies were applied for 2 h, followed by two washing steps. Hoechst33342 was used for nuclear staining at 1 μg/mL for 30 min in PBS. The images were taken using a x40 objective. CD68 was used as a macrophage and lysosomal marker. The experiment was imaged using an EVOS FL Auto fluorescence microscope (Thermo Fisher Scientific, Waltham, MA, USA). The acquired images were processed using ImageJ (Wayne Rusband; National Institute of Health, Bethesda, MD, USA).

For the immunocytochemistry, VP1 VLP-specific murine MAbs against recombinant PyV-derived VP1 VLPs were used (#MAb clone—polyomavirus indicated): #8G8—JCPyV; #5G8—KIPyV; #12F8—WUPyV, #9G6—MCPyV; #25F3—HPyV6; #1H1—HPyV7, #16H6—TSPyV; #18G5—HPyV9; #14A12—STLPyV (Appendix A).

### 4.11. Statistical Analysis

All statistical analyses were performed with GraphPad Prism 9.2.0 (GraphPad Software, Inc., La Jolla, CA, USA). The data in the figures are represented as individual data points from at least 3 independent experiments. Independent experiments referred to as *n* means the number of mice. Antibody titres were statistically compared using two-way ANOVA in conjunction with Šídák’s multiple comparisons test. Differences with a *p* value less than 0.05 were considered to be statistically significant: * *p* < 0.05, ** *p* < 0.01, *** *p* < 0.001, **** *p* < 0.0001.

## Figures and Tables

**Figure 1 ijms-24-04907-f001:**
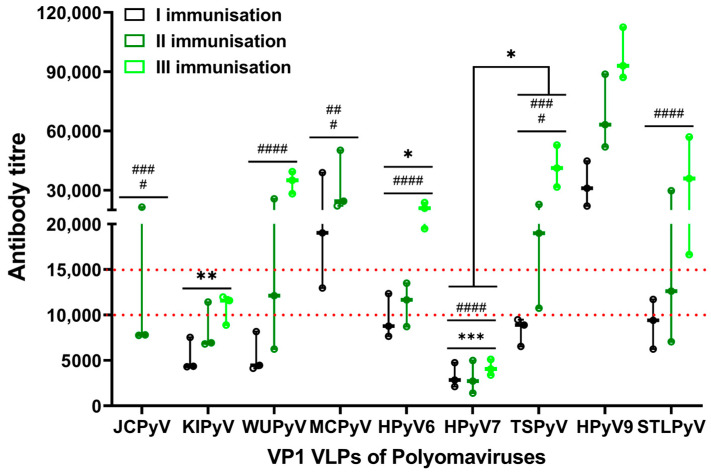
Titres of VP1-specific IgG in the antisera of mice immunised with different HPyV VP1 VLPs. The microtiter plate was coated with various recombinant VP1 VLPs and subsequently incubated with antiserum samples collected on the 14th day after immunisation. Antiserum titres after the first, second, and third (for MCPyV VP1 VLPs after the first and second, and for JCPyV VP1 VLPs after the second) immunisation as one data unit was statistically compared between different polyomavirus VP1 VLPs. Significantly different pairs were detected, with MCPyV, TSPyV and HPyV9 VP1 VLPs groups. Pairs with MCPyV VP1 VLPs are shown with *. Pairs with HPyV9 VP1 VLPs are shown with #. Three mice were dedicated per group for immunisation. Data are represented using box-plots, *^/#^ *p* < 0.05, **^/##^ *p* < 0.01, ***^/###^
*p* < 0.001, ^####^
*p* < 0.0001 according to two-way ANOVA followed by Šídák’s multiple comparisons test.

**Figure 2 ijms-24-04907-f002:**
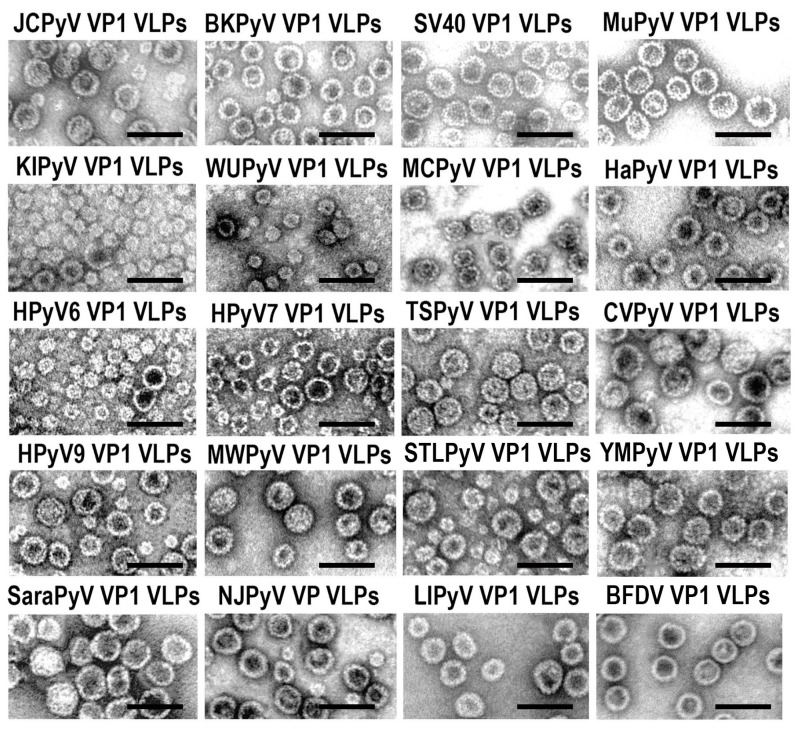
Electron-microscopical images of the recombinant PyV VP1-derived VLPs produced in yeast *Saccharomyces cerevisiae* that were used in this study. The scale bar (black line) represents 100 nm.

**Figure 3 ijms-24-04907-f003:**
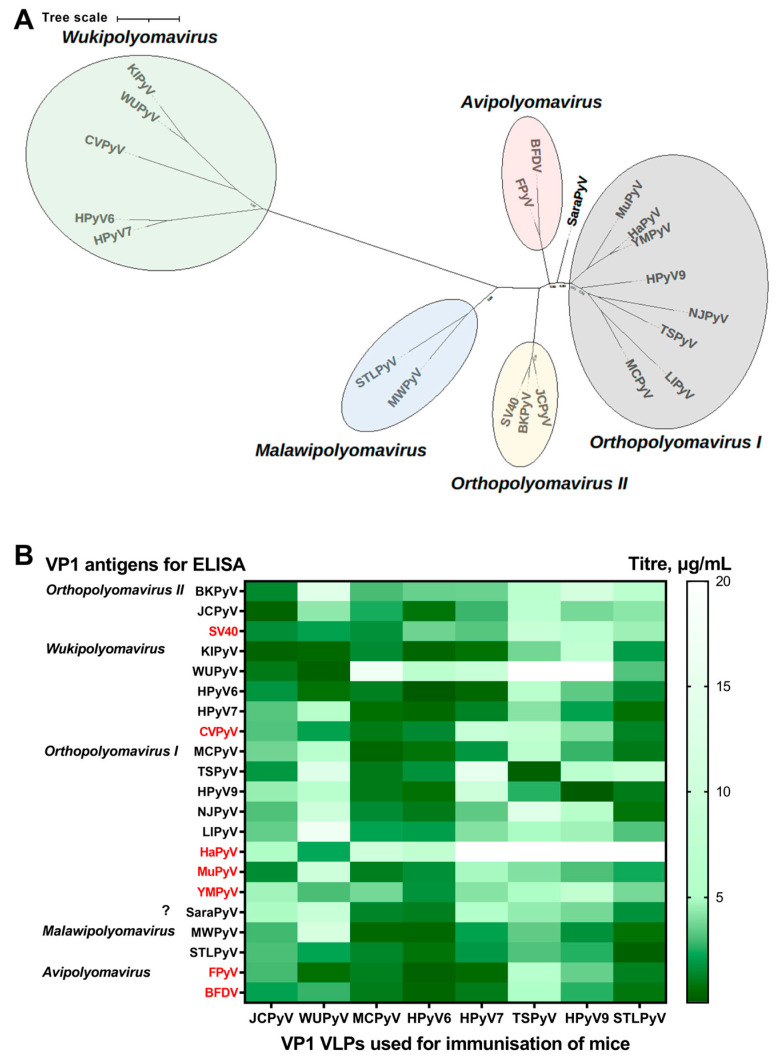
Cross-reactivity of mouse antisera with VP1 proteins of different PyVs. (**A**). Unrooted maximum likelihood phylogenetic tree of PyVs based on VP1 amino acid sequences. Five distinct phylogenetic groups: *Wukipolyomaviru*s, *Avipolyomavirus*, *Orthopolyomavirus I* and *II*, and *Malawipolyomavirus,* are marked in different colors. Transfer bootstrap expectation values lower than 0.9 are shown. (**B**). Antisera raised against VP1-derived VLPs of different HPyVs were tested with different VP1 VLPs of mammalian, including human, and avian PyVs by an indirect ELISA. The diluent buffer was used as a negative control. The X axis presents the antigens used for immunization of mice, while the Y axis shows antigens used for the coating of ELISA plates. On the left panel, PyVs are grouped according to similarities of their VP1 protein sequence. Non-human PyVs are highlighted in red. The smaller values of titres in the heat map mean a stronger antibody reactivity. The values of titres (µg/mL) in the heat map equal to 20 are assumed as showing no cross-reactivity according to the negative control; 5–15—moderate reactivity; 1–5—strong reactivity; <1—very strong reactivity. Titration curves are shown in Appendix A.

**Figure 4 ijms-24-04907-f004:**
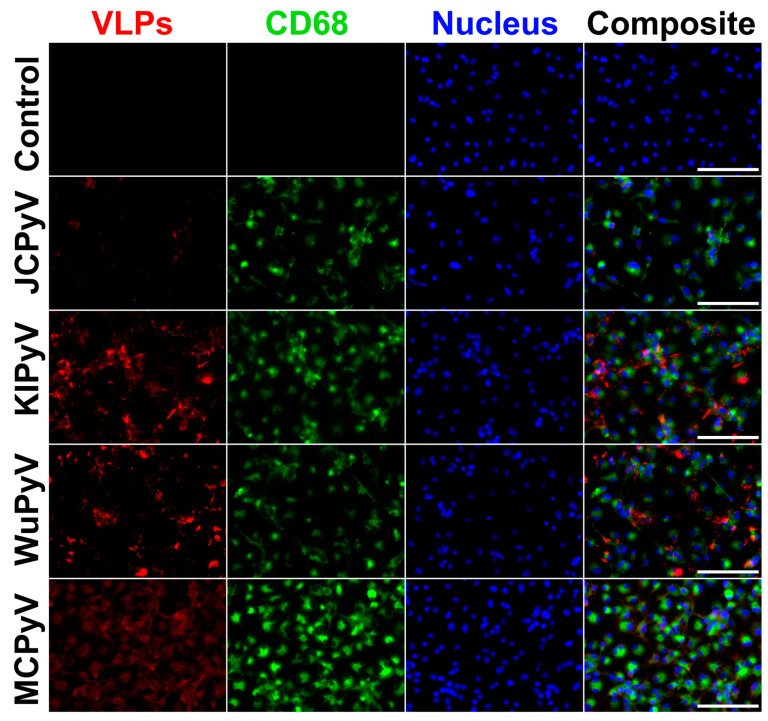
The uptake of VP1 VLPs of different HPyVs by primary mouse macrophages. Macrophages were treated with VP1 VLPs for 24 h at 20 µg/mL concentration. Cells were immunostained with VP1 VLP-specific MAbs (red), anti-CD68—macrophage and lysosomal marker (green), nuclear stain Hoechst33342 (blue) and analysed by fluorescence microscopy. The experiment was repeated twice, and representative images of one experiment are shown. Secondary antibody control is referred to as control. Images were taken using ×40 objective. The scale bars indicate 100 μm.

**Table 1 ijms-24-04907-t001:** The list of PyVs for which VP1-derived VLPs were used in the study.

PyV Name	Abbreviation	PyV Species	VP1 Sequence ID	Reference
BK polyomavirus	BKPyV	*Human polyomavirus 1* (HPyV1)	AFA41874.1	[17]
John Cunningham polyomavirus	JCPyV	*Human polyomavirus 2* (HPyV2)	AAB84311.1	[17]
Karolinska Institute polyomavirus	KIPyV	*Human polyomavirus 3* (HPyV3)	YP_001111258.1	[18]
Washington University polyomavirus	WUPyV	*Human polyomavirus 4* (HPyV4)	YP_001285487.1	[18]
Merkel cell polyomavirus	MCPyV	*Human polyomavirus 5* (HPyV5)	AHW79944.1	[18]
Human polyomavirus 6	HPyV6	*Human polyomavirus 6* (HPyV6)	YP_003848918.1	[18]
Human polyomavirus 7	HPyV7	*Human polyomavirus 7* (HPyV7)	ADE45454.1	[18]
Trichodysplasia spinulosa-associated polyomavirus	TSPyV	*Human polyomavirus 8* (HPyV8)	YP_003800006.1	[18]
Human polyomavirus 9	HPyV9	*Human polyomavirus 9* (HPyV9)	YP_004243705.1	[18]
Malawi polyomavirus	MWPyV	*Human polyomavirus 10* (HPyV10)	AFN02459.1	[18]
Saint Louis polyomavirus	STLPyV	*Human polyomavirus 11* (HPyV11)	YP_007354884.1	[18]
Human polyomavirus 12 (HPyV12)	SaraPyV	*Sorex araneus polyomavirus 1*	YP_007684355.2	[18]
New Jersey polyomavirus	NJPyV	*Human polyomavirus 13* (HPyV13)	YP_009030020.1	[18]
Lyon-IARC polyomavirus	LIPyV	*Human polyomavirus 14* (HPyV14)	YP_009352870.1	This study
Simian virus 40	SV40	*Macaca mulatta polyomavirus 1*	P03087.2	[17]
Hamster polyomavirus	HaPyV	*Mesocricetus auratus polyomavirus 1*	YP_009111410.1	[30]
Murine polyomavirus	MuPyV	*Mus musculus polyomavirus 1*	P49302.2	[17]
Common vole polyomavirus	CVPyV	*Microtus arvalis polyomavirus 1*	YP_009174987.1	[31]
Yellow-necked mouse	YMPyV	*Apodemus flavicollis polyomavirus 1*	AWD33708.1	This study
Budgerigar fledgling disease virus	BFDV	*Aves polyomavirus 1*	YP_004061428.1	[17]
Finch polyomavirus	FPyV	*Pyrrhula pyrrhula polyomavirus 1*	YP_529833.1	[32]

## Data Availability

The data used to support the findings of this study are included within the article. All other data supporting the findings of this study will be made available upon reasonable request to the corresponding authors.

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
