# Peer review of "Immunogenic Properties and Antigenic Similarity of Virus-like Particles Derived from Human Polyomaviruses"

_ijms, 2023, doi:10.3390/ijms24054907_

Round 1

Reviewer 1 Report

The manuscript “Immunogenic properties and antigenic similarity of virus-like particles derived from human polyomaviruses” is well designed and has a significant novelty in the viral vaccine adjuvants process. The manuscript acceptable for publication with some minor edits.

What is VP1 in line#13.

Write the methods used to study the immunogenicity and antigenic properties of VP1 VLPs in the research.

Define BK in line# 29.

Include the scale bar size in the Figure 2 microscopic images. Cite the figure number in the text.

MAbs could be used as a highly specific tool for PyV detection and characterisation. How to detect from the figure 2 not discussed.

Author Response

Answer to comment "What is VP1 in line#13.": viral protein 1, changed in manuscript line 13 to "viral protein 1 (VP1)"

Answer to comment "Write the methods used to study the immunogenicity and antigenic properties of VP1 VLPs in the research.": the methods were supplemented in lines 497-514:

The study of immunogenicity and antigenic properties of VP1 VLPs

Antiserum samples collected from immunised mice were used for the immunogenicity study. Equal dilution factors of polyclonal antibodies were used to generate titration curves. Titres were determined with GraphPad Prism 9.2.0 (GraphPad Software, Inc., La Jolla, CA) by applying Sigmoidal, 4PL, X is log(concentration) model and calculating EC50 (half maximal effective concentration). For each investigated antigen 3 biological experiments were used (polyclonal antibodies from 3 mice). For each antigen 1st, 2nd and 3rd immunisations were grouped as one data set. Antibody titres data sets were compared with each other using two‐way ANOVA. The antigenicity is referred to antibody specificity to antigen. It was investigated by analysing the antibodies' crossreactivity with antigens of different PyV VP1. Equal concentrations (µg/ml) of polyclonal antibodies were used to generate titration curves. Antiserum concentration was equated to the total protein concentration. Titres were determined with GraphPad Prism 9.2.0 (GraphPad Software, Inc., La Jolla, CA) by applying Sigmoidal, 4PL, X is log(concentration) model and calculating EC50. For each investigated antibody 2 technical replicates were used. Calculated titres were laid out in the heatmap for comparison of each polyclonal antibody reactivity with different VP1 VLPs of human and animal PyVs.

Answer to comment "Define BK in line# 29.": The BK virus cannot be defined in more detail because it was first isolated in 1971 from the urine of a renal transplant patient with the initials B.K. and named BK (Gardner SD, Field AM, Coleman DV, Hulme B (June 1971). "New human papovavirus (B.K.) isolated from urine after renal transplantation". Lancet. 1 (7712): 1253–7. doi:10.1016/s0140-6736(71)91776-4).

Answer to comment "Include the scale bar size in the Figure 2 microscopic images. Cite the figure number in the text.": the changes were done in manuscript line 176. Figure 2 is cited in line 152.

Answer to comment "MAbs could be used as a highly specific tool for PyV detection and characterisation. How to detect from the figure 2 not discussed.": in manuscript lines 363-367 the text about the use of MAbs was added: 

In this study we also generated a collection of MAbs of various specificities to HPyVs. This tool can be used for PyVs detection and characterisation. Generated VP1 specific MAbs could be applied for assays to detect the native viruses in cell cultures or tissues. MAbs specificity to a certain epitope may also be used to investigate the structure of viral particles and interactions between those particles and host cell receptors.

Reviewer 2 Report

                This study builds upon some older work published by the group on immune responses to polyomavirus VP1 VLPs generated in yeast and extends similar observations to 9 other human polyomaviruses.  The experimental design is straightforward and the conclusions are fully supported by the results.  While the study results largely confirm and support previous data or expectations given the evolutionary similarity of the various polyomaviruses, nevertheless the study represents findings that will be of interest to specialists in the field.  I only have a few suggestions to polish the presentation:

Major Point

1.        Fig. 1:  Given the surprisingly distinct failure of HPyV7 to elicit a similar antibody response as the others, it would perhaps be useful if the authors would provide some evidence on the quality control for the VLPs that were produced (in addition to the TEM pictures in Fig. 2 that demonstrate size heterogeneity) to determine if they were all of comparable stability.

Minor Points:

1.        Introduction:  The writing can be sharpened in some places in the manuscript.  For example, change lines 32 to 34 from

Both viruses enter the body via respiratory system and can also cause mild respiratory infection [5]. Only later, starting 2007 additional twelve human polyomaviruses (HPyVs) were identified.

                      To

 Both viruses enter the body via the respiratory system and can also cause mild respiratory infection [5]. Only later, starting in 2007, an additional twelve human polyomaviruses (HPyVs) were identified.

2.       Line 236:  The apparently spurious statement:  This is example 1 of an equation:  should be removed.  The text of the results section is also small (similar to the legend text) and needs to be re-sized.

3.       Figure 4:  While representative images of one experiment are shown, the authors should also explicitly state how many experiments this is representative of (e.g. how many times was the experiment repeated) to establish rigor in the conclusions.

Author Response

Point 1 "Fig. 1:  Given the surprisingly distinct failure of HPyV7 to elicit a similar antibody response as the others, it would perhaps be useful if the authors would provide some evidence on the quality control for the VLPs that were produced (in addition to the TEM pictures in Fig. 2 that demonstrate size heterogeneity) to determine if they were all of comparable stability."

Answer1: the answer is in the uploaded file.

Point 2: "Introduction:  The writing can be sharpened in some places in the manuscript.  For example, change lines 32 to 34 from

Both viruses enter the body via respiratory system and can also cause mild respiratory infection [5]. Only later, starting 2007 additional twelve human polyomaviruses (HPyVs) were identified.

                      To

 Both viruses enter the body via the respiratory system and can also cause mild respiratory infection [5]. Only later, starting in 2007, an additional twelve human polyomaviruses (HPyVs) were identified."

Answer2:  The change was done in the manuscript file.

Point 3: Line 236:  The apparently spurious statement:  This is example 1 of an equation:  should be removed.  The text of the results section is also small (similar to the legend text) and needs to be re-sized.

Aswer3: "This is example 1 of an equation" was deleted and the section was resized in the manuscript file.

Point 4: Figure 4:  While representative images of one experiment are shown, the authors should also explicitly state how many experiments this is representative of (e.g. how many times was the experiment repeated) to establish rigor in the conclusions.

Answer4: the figure legend was supplemented with bold text in the manuscript file:

The uptake of VP1 VLPs of different HPyVs by primary mouse macrophages. Macrophages were treated with VP1 VLPs for 24 h at 20 µg/ml concentration. Cells were immunostained with VP1 VLP-specific MAbs (red), anti-CD68 – macrophage and lysosomal marker (green), nuclear stain Hoechst33342 (blue) and analysed by fluorescence microscopy. The experiment was performed twice and representative images of one experiment are shown. Secondary antibody control is referred as control. Images were taken using x40 objective. The scale bars indicate 100 μm.
